# Jamming and Anti-Jamming Strategies of Mobile Vehicles

Gleb Dubosarskii *[ID] and Serguei Primak

Department Electrical and Computer Engineering, Western University, London, ON N6A 5B9, Canada; slprimak@uwo.ca
* Correspondence: gdubosar@uwo.ca

**Abstract:** Anti-jamming games have become a popular research topic. However, there are not many publications devoted to such games in the case of vehicular ad hoc networks (VANETs). We considered a VANET anti-jamming game on the road using a realistic driving model. Further, we assumed the quadratic power function in both vehicle and jammer utility functions instead of the standard linear term. This makes the game model more realistic. Using mathematical methods, we expressed the Nash equilibrium through the system parameters in single-channel and multi-channel cases. Since the network parameters are usually unknown, we also compared the performance of several reinforcement learning algorithms that iteratively converge to the Nash equilibrium predicted analytically without having any information about the environment in the static and dynamic scenarios.

**Keywords:** anti-jamming game; communication game; reinforcement learning

## 1. Introduction

Vehicular ad hoc networks (VANETs) are designed to provide communication between vehicles, as well as between vehicles and infrastructure. Similar to any other networks, they are vulnerable to jammer attacks that try to disrupt communications between vehicles. To achieve this goal, the jammer creates a denial-of-service (DoS) attack by sending fake or replayed signals in order to flood the network with traffic. Such attacks are an important problem for which an effective solution must be developed. To combat jammer attacks, the vehicles can change transmission channels, increase transmission power, or even change their location to avoid being in close proximity to the jammer. It is convenient to consider such a game in terms of game theory [1]. Within the framework of this theory, it is assumed that one or more cooperating vehicles maximize their utility functions, while each of the jammers maximizes its own. In the simplest case, the vehicle utility function is the difference between the signal-to-interference-plus-noise ratio (SINR) and signal transmission power with a certain coefficient. In other words, the goal of the vehicles is to maximize SINR under the assumption of reasonable signal transmission power. The jammer utility function is defined as the difference of the vehicle utility function taken with the opposite sign and its signal transmission power with a different coefficient. This means that the jammer minimizes the vehicle utility function and has limitations on its signal transmission power. The optimal strategy of the vehicles and the jammers can be determined as the Nash equilibrium of this game.

However, the game theory approach cannot always be used explicitly in practice, since the network parameters are usually unknown. Because of this, over the past decade, researchers have been finding the optimal solution implicitly using machine learning algorithms [2], which has become popular in research. These algorithms, through trial and error, constantly improve their strategies. One of the most popular algorithms is Q-learning [3] and its modifications, due to their fast convergence and simplicity of implementation. We discuss below several articles that use game theory and machine learning algorithms for finding optimal strategies in the anti-jamming game in various settings.

In [4], the authors examine a case in which the jammer attempts to disrupt the communication between two consecutive vehicles in a platoon. Since data are transmitted over one channel, both the jammer and the vehicles adapt their transmission power in order to maximize their utility functions. The authors use a modification of Q-learning algorithm called Dyna-Q and compare the learning outcomes with the classic Q-learning algorithm. The results convincingly show that Dyna-Q converges to the same strategy, but has a higher convergence rate. The articles [5,6] discuss the VANET anti-jamming game with drones. The essence of the game is that in the case of a jammer attack, the drone replays the data sent by VANET nodes in order to increase SINR and reduce bit error ratio (BER). Thus, at every time moment, the drone decides whether to send data or not. The authors deduce the Nash equilibrium [7] and compare it with results iteratively found by Q-learning and its effective modification called policy hill climbing.

The case of a cooperative game between devices is also popular in the literature. The difference between this case and single-agent games is that instead of having independent utility functions, devices have a common utility function that evaluates the network state as a whole. Such games are more difficult to consider, since in this case there is a large number of system states that grow exponentially with the device number. In [8,9], the authors examine an anti-jamming game with several transmitting cooperating devices. Q-learning algorithm for finding a common optimal strategy in this formulation shows its advantage over the non-cognitive sub-band selection policy. In [10], a game with cooperating devices and one jammer is considered. The authors propose an iterative algorithm for finding a cooperative strategy and compare the results with random anti-jamming and selfish anti-jamming algorithms. The simulation results show that the iterative algorithm achieves higher throughput and better performance than non-cooperative algorithms.

Recent articles consider satellite anti-jamming games. In [11], the authors discuss the anti-jamming coalition game. The purpose of the satellites is to form subnetworks in order to transmit information under jamming attacks with a minimum energy consumption, while the task of the jammer is to find the optimal location. The authors successfully use Q-learning algorithm in order to find a suboptimal satellite strategy in an unknown jamming environment. In [12], the authors apply deep reinforcement learning to find a satellite routing scheme and a fast response anti-jamming algorithm.

In this article, we examine a VANET anti-jamming game as in [4–6]. However, instead of the static model considered in [4], we use the intelligent driver model [13] to describe the evolution of the network in the case in which the vehicles are organized into a platoon. All simulations were carried out in the case of a straight road. We assume that the two communicating vehicles are pursued by the jammer interrupting their ongoing communication. We considered two cases: a single-channel and multi-channel game. In the multi-channel case, it is assumed that the vehicles change channels according to a predetermined pseudo-random sequence. In this situation, we presume that the jammer shares its power between channels because it cannot predict the next state of the network in advance. To confirm the simulation results, we formulate and prove theorems that describe the Nash equilibrium of the game, which can be interpreted as the optimal strategy for the vehicle and the jammer. At first, we suggest that power included in the vehicle and jammer utility functions is linear, but from the Theorem 1 presented in this paper it follows that the optimal vehicle strategy is to transmit at maximum power; therefore, we change the classic formula of the utility function in order to find a non-trivial vehicle strategy. To do this, we consider the quadratic power function in both vehicle and jammer utility functions. Such a power function is closer to practical implementation, since transmitting on higher power levels requires a greater expenditure of system resources than at low levels. Under this assumption, we formulate and prove the Nash equilibrium theorems in both single-channel and multi-channel cases. Next, we examined several machine learning algorithms such as policy hill climbing, deep Q-learning, dueling Q-learning, and dueling deep Q-learning. All the algorithms successfully converge to the theoretically derived Nash equilibrium. To the best of our knowledge, this is the first article to discuss an anti-jamming

game with a quadratic power function and analytical derivation of the Nash equilibrium in the multi-channel case. We also want to note that this article makes a comprehensive comparison of modern Q-learning algorithms.

The article is organized as follows. In Section 2, the anti-jamming game is described in terms of game theory. In Sections 3 and 4, the necessary and sufficient conditions of the Nash equilibrium are established in the single-channel and multi-channel cases. Section 5 describes machine learning algorithms that are used in Section 6 in order to find the optimal vehicle and jammer strategies. We provide a diagram explaining the structure of the article in Figure 1.

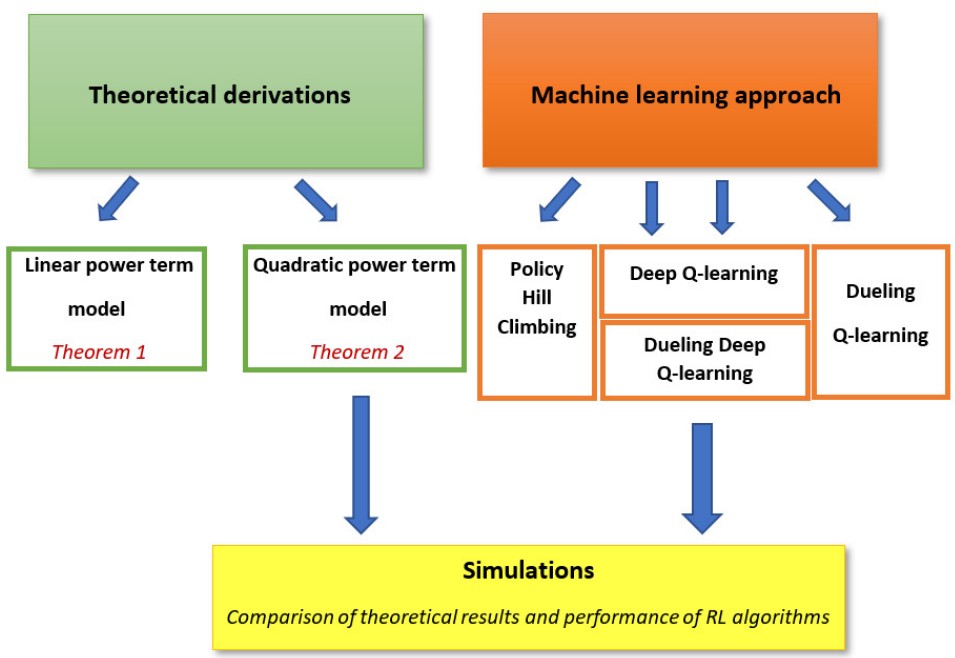

**Figure 1.** Content and structure of the article.

## 2. Game Description

In this section we consider the anti-jamming game on a single road. We assume that two communicating cars are chased by the jammer. Signal-to-interference-plus-noise ratio (SINR) of the vehicle is given by the formula

$$SINR = \frac{h_{car}^2 x}{\sigma^2 + h_J^2 y},$$

where $\sigma^2$ is a noise power, $h_{car}^2$ and $h_J^2$ are vehicle and jammer channel power gains, $x$ and $y$ are signal transmission powers of the car and the jammer, respectively (see Figure 2).

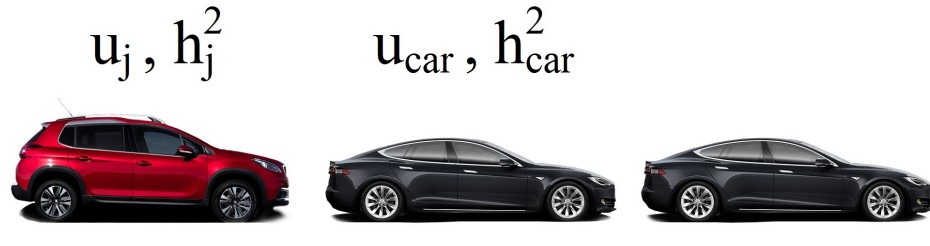

**Figure 2.** Illustration of anti-jamming game.

We assume that the utility functions of the vehicle $u_{car}$ and the jammer $u_J$ are calculated by the formulas taken from [14]

$$u_{car} = \frac{h_{car}^2 x}{\sigma^2 + h_J^2 y} - C_{car} x, \tag{1}$$

$$u_J = -\frac{h_{car}^2 x}{\sigma^2 + h_J^2 y} + C_{car} x - C_J y, \tag{2}$$

where $C_{car}$ and $C_J$ are positive transmission costs of the vehicle and the jammer. The goal of each player is to maximize their utility functions. In the next section, we consider utility functions with quadratic power function, but in this section we focus on this standard model.

We assume that after the signal is transmitted, the vehicle receives back the signal-to-interference-plus-noise ratio (SINR) value and, based on this information, makes a decision on increasing/decreasing the transmission power. By $P_C$ and $P_J$ we denote maximum vehicle and jammer transmission power, respectively.

### 3. Nash Equilibrium in the Case of the Linear Cost Function

By definition, the Nash equilibrium is a strategy $(x^*, y^*)$, which satisfies the following two inequalities:

$$u_{car}(x^*, y^*) \geq u_{car}(x, y^*), \tag{3}$$

$$u_J(x^*, y^*) \geq u_J(x^*, y). \tag{4}$$

**Theorem 1.** *1. If the inequality*

$$C_{car} \geq \frac{h_{car}^2}{\sigma^2}$$

*holds then the point* $(0,0)$ *is a Nash equilibrium.*
*2. If the opposite inequality*

$$C_{car} < \frac{h_{car}^2}{\sigma^2}$$

*is satisfied then the Nash equilibrium is reached at the point* $(x^*, y^*)$ *that can be expressed in terms of the parameter*

$$\hat{y} = \frac{-\sigma^2}{h_J^2} + \frac{h_{car}}{h_J} \sqrt{\frac{P_J}{C_J}}$$

*as follows:*

$$x^* = P_C,$$

$$y^* = \begin{cases} \hat{y}, & \text{if } 0 \le \hat{y} \le P_J, \\ 0, & \text{if } \hat{y} < 0, \\ P_J, & \text{if } \hat{y} > P_J. \end{cases} \tag{5}$$

**Remark 1.** *As can be seen from Theorem 1, the optimal vehicle strategy is trivial. The vehicle must transmit at maximum power or zero power, depending on the parameter values. Theorem 1 is proved in Appendix A.*

### 4. Nash Equilibrium in the Case of the Quadratic Cost Function

In the previous section, we examine the case in which the power increases linearly in the car and jammer utility functions. Since the linear power term leads to a trivial car strategy (see Theorem 1 and the remark), we decide to consider the quadratic power term in the vehicle and jammer utility functions. In addition, it better reflects the real game scenario, because the transmission cost grows faster than linearly with transmission power increase.

In the case of a multi-channel game with $m$ channels, we assume that the vehicle selects the next channel according to a predetermined pseudorandom sequence, and all channels are chosen equally probable. We assume that the jammer divides its power between $m$ channels. Thus, the jammer state is the vector $(y_1, y_2, \ldots, y_m)$, where $y_i$ is the power transmitted through the channel $i$. Since the vehicle chooses a certain channel with probability $1/m$ and the vehicle utility function on this channel is given by Formula (1), we conclude that the average value of the received reward is calculated by the formula

$$u_{car} = \sum_{k=1}^{m} \frac{1}{m} \left( \frac{h_{car}^2 x}{\sigma^2 + h_J^2 y_k} - C_{car} x^2 \right). \tag{6}$$

We assume that the jammer utility function is given by the following formula:

$$u_J = -u_{car} - C_J \sum_{k=1}^{m} y_k^2 = \sum_{k=1}^{m} \frac{1}{m} \left( -\frac{h_{car}^2 x}{\sigma^2 + h_J^2 y_k} + C_{car} x^2 \right) - C_J \sum_{k=1}^{m} y_k^2. \tag{7}$$

The sum of the transmitted jammer powers $y_k$ over all $k$ channels must not exceed the maximum jammer transmission power $P_J$ and should be positive

$$\sum_k y_k \le P_J, \tag{8}$$

$$y_k > 0. \tag{9}$$

In addition, we assume that the car power $x$ lies in the following range:

$$0 < x \le P_C. \tag{10}$$

The point $(x^*, y_1^*, y_2^*, \ldots, y_m^*)$ is called a Nash equilibrium if the following inequalities:

$$u_{car}(x^*, y_1^*, y_2^*, \ldots, y_m^*) \ge u_{car}(x, y_1^*, y_2^*, \ldots, y_m^*), \tag{11}$$

$$u_J(x^*, y_1^*, y_2^*, \ldots, y_m^*) \ge u_J(x^*, y_1, y_2, \ldots, y_m) \tag{12}$$

are satisfied.

**Theorem 2.** *Nash equilibrium $(x^*, y_1^*, y_2^*, \ldots, y_m^*)$ exists in the game with cost functions (6), (7) in the region given by inequalities (8)–(10) if and only if*

$$y_1^* = y_2^* = \ldots = y_m^* = y^* \tag{13}$$

*and one of the following conditions is satisfied.*

**1.** $0 < x^* < P_C$, $0 < y^* < \frac{P_J}{m}$. In this case, $y^*$ can be found from the equation

$$y^*(\sigma^2 + h_J^2 y^*)^3 = \frac{h_{car}^4 h_J^2}{4mC_{car}C_J}. \tag{14}$$

*The value of $x^*$ is expressed in terms of $y^*$ according to the formula*

$$x^* = \frac{h_{car}^2}{2C_{car}(\sigma^2 + h_J^2 y^*)}. \tag{15}$$

**2.** $x^* = P_C$, $0 < y^* < \frac{P_J}{m}$. *In this case $y^*$ is a solution to the equation*

$$y^*(\sigma^2 + h_J^2 y^*)^2 = \frac{h_{car}^2 h_J^2 P_C}{2mC_J} \tag{16}$$

*and the inequality*

$$\frac{h_{car}^2}{\sigma^2 + h_J^2 y^*} - 2C_{car}P_C \geq 0 \tag{17}$$

*holds.*

**3.** $0 < x^* < P_C$, $y^* = \frac{P_J}{m}$. *The value $x^*$ is given by the formula*

$$x^* = \frac{h_{car}^2}{2C_{car}(\sigma^2 + h_J^2 \frac{P_J}{m})}, \tag{18}$$

*and the following inequality:*

$$\frac{h_{car}^2 h_J^2 x^*}{(\sigma^2 + h_J^2 \frac{P_J}{m})^2} - 2C_J P_J \geq 0 \tag{19}$$

*is satisfied.*

**4.** $x^* = P_C$, $y^* = \frac{P_J}{m}$. *In this case, the following conditions must be satisfied:*

$$\frac{h_{car}^2}{\sigma^2 + h_J^2 \frac{P_J}{m}} - 2C_{car}P_C \geq 0, \tag{20}$$

$$\frac{h_{car}^2 h_J^2 P_C}{(\sigma^2 + h_J^2 \frac{P_J}{m})^2} - 2C_J P_J \geq 0. \tag{21}$$

The proof of Theorem 2 is given in Appendix B.

**Remark 2.** *The case of one channel is a special case of the case of several channels. To consider this case, it is enough to substitute $m = 1$ into Formulas (6)–(12) and Theorem 2.*

**Remark 3.** *Equalities (14) and (16) are equations for the variable $y^*$, which can be solved numerically using binary search, since their left side is an increasing function, and the right side is a constant.*

## 5. Machine Learning Solution

In the previous section we derive an analytical expression for the Nash equilibrium in the case of multi-channel and single-channel games. However, communication parameters such as channel gains are generally unknown, which limits practical applications of these results. Therefore, in practice, it is of interest to use machine learning algorithms, which by trial and error find the optimal strategy. In this section, we compare the performance of policy hill climbing [15] and several state-of-the-art modifications of the classic Q-learning

algorithm [3]. All these algorithms are general and can be applied to turn-based games. The essence of the game is that at each step of $k$ each agent is in some state $s_k$ and at each turn performs the action $a_k$ for which the agent receives a reward $r_k$. The goal of the agent is to maximize cumulative reward

$$\mathbf{E}\Big\{\sum_k \gamma^k r_k\Big\}, \tag{22}$$

where $\mathbf{E}$ is a mathematical expectation and $\gamma$ is called a discount-rate. Maximizing value (22) could be interpreted as maximizing the average cumulative reward obtained by following a certain probabilistic policy. The discount rate $\gamma, (0 < \gamma < 1)$ determines how important the future rewards are to the agent. If $\gamma$ is close to 1, this means the high importance of the rewards, otherwise they are less important and the agent focuses more on the current state.

The essence of classic Q-learning algorithm is in recalculating the Q-matrix. The value $Q[s, a]$ measures the quality of being in the state $s$ and performing the action $a$. With a probability $\varepsilon$, the agent selects an action randomly; in other cases it acts greedily and selects an action with a maximum Q-value. Often the parameter $\varepsilon$ decreases to zero with an increasing number of iterations, since it is believed that the environment becomes explored quite well over time and does not require future exploration. In the article, we use exponential $\varepsilon$ decay with a starting value of $\varepsilon_0$, a limiting value of $\varepsilon_\infty$, and *decay_rate* given by the formula

$$\varepsilon = \varepsilon_\infty + (\varepsilon_0 - \varepsilon_\infty)e^{-iteration\_number/decay\_rate}, \tag{23}$$

where *iteration_number* is the number of the game iterations until the current moment. At each step, the recalculation of Q-matrix is performed according to the formula

$$Q[s_k, a_k] = Q[s_k, a_k] + \alpha(r_k + \gamma \max_a(Q[s_{k+1}, a] - Q[s_k, a_k]). \tag{24}$$

One of the main parameters of this algorithm is learning rate $\alpha$ which determines how significant the impact of new experience on Q-values would be.

Policy hill climbing (PHC) is a modification of Q-learning. Its main difference from Q-learning is the choice of action. Q-learning is based on the greedy choice of the action $a$ from the state $s$ with the highest value $Q[s, a]$ and exploration of the environment with the probability $\varepsilon$. PHC selects each action with a certain probability, which is updated each time taking into account the received rewards.

Modifications of the classical Q-learning algorithm [3] discussed below are based on neural networks. A feature of deep Q-learning [16] is that instead of memorizing values in Q-matrix (the size of which could be very large), the algorithm trains the neural network to store Q-values. To do this, it uses a special method of training, called experience replay. The essence of this method is that previous experiences are stored in a buffer of constant length *replay_mem_size* and are repeatedly used to train a neural network. After each iteration of the algorithm, *batch_size* of the previous experiences are randomly extracted from the buffer and used to retrain the network.

Another modification of Q-learning is called double Q-learning [17]. Its creation is caused by the fact that values of Q-matrix can be locally overestimated in comparison with the real values. The essence of this modification is that instead of using one network, two are used. One is the current version of the network, and the other is an old copy saved a few steps back. An old copy of the network is updated every *update_target_frequency* iterations. One network version is used for value evaluation and another for the next action selection.

A modification called dueling Q-learning [18] improves convergence and stabilizes the training process by introducing a new element called advantage. The essence of advantage is that it is used to compare the Q-value of the current action and the average Q-value, so the algorithm tries to encourage more promising actions. We also implement

double dueling Q-learning, which is a combination of the ideas of double Q-learning and dueling Q-learning.

The anti-jamming game algorithm is described in pseudo-code (Algorithm 1). We first consider the case of Q-learning modifications based on neural networks such as deep Q-learning, dueling Q-learning, and dueling double Q-learning, and then describe what should be changed in this pseudo-code if PHC is used. We discretize the range $[0, P_J]$ into *power_level_num* levels. In line 2, by exhaustive search over all power distributions the jammer finds an optimal distribution of power between $m$ channels. Taking into account the previous vehicle state in line 3 the vehicle locations are updated using the intelligent driver model from [13]. In line 4, the transmitting vehicle changes the channel according to the pseudo-random sequence, which is assumed to be unknown to the jammer. To discretize the SINR in lines 5 and 8, the *disc_step* step size is used. Line 6 evaluates the value of $\varepsilon$ which is responsible for the amount of exploration in Q-learning algorithms. Regardless of which modification of the Q-learning algorithm we use lines 7–10, which look the same. In line 7, the algorithm predicts the next action. To do this, it returns a value of 2, 1, or 0, meaning that the vehicle must transmit signal on a higher power level, stay at the current level, or go to a level lower, respectively. It must be ensured that the level does not go beyond the permissible power limits. In line 9, the new system state is added to memory as an array of four values $(SINR\_old, new\_action, SINR\_new, reward)$. Line 10 calls a function that retrieves the *batch_size* of previous experiences, updates the state estimate using Formula (24), and trains the neural network to remember the updated values. In the case of PHC, line 6 is not needed, since this algorithm does not have the parameter $\varepsilon$, in line 10, the called algorithm additionally recalculates the probabilities with which actions would be selected in the future.

---

**Algorithm 1** Anti-jamming game algorithm

---

1: **while** (Game is not terminated) **do**
2:　　　Recalculate jammer power distribution
3:　　　Recalculate state of the system using intelligent Driver model
4:　　　Choose new vehicle transmission channel according to pseudo-random sequence
5:　　　Retrieve and discretize $SINR\_old$ from memory obtained from the previous iteration
6:　　　Calculate $\varepsilon$ using exponential decay rule (23)
7:　　　$new\_action = Learning\_Algorithm(SINR\_old, \varepsilon)$
8:　　　Calculate and discretize $SINR\_new$ after action $new\_action$
9:　　　Add to memory $(SINR\_old, new\_action, SINR\_new, reward)$
10:　　　Retrain algorithm
11:　　　Save current state of the system
12: **end while**

---

## 6. Simulations

We consider the case of adaptive jammer, which is the most dangerous for the network. Based on the state of communication at the previous moment, the jammer finds the optimal transmit power by considering all the possible options. We performed simulations in the single-channel and multi-channel cases and compared the performance of PHC, deep Q-learning, and its recent modifications dueling Q-learning and dueling double Q-learning. We assume the following parameter values (see their description in the previous section):

$$\gamma = 0.7, replay\_mem\_size = 50, batch\_size = 32,$$

$$update\_target\_frequency = 20, \varepsilon_0 = 1,$$

$$\varepsilon_\infty = 0.01, decay\_rate = 100, disc\_step = 0.05.$$

We use the intelligent driver model [13] to generate car motion. According to it, the movement of vehicle $i$ on a single road is described by the following differential equations:

$$\dot{x}_i = v_i, \tag{25}$$

$$\dot{v}_i = a \left( 1 - \left( \frac{v_i}{v_0} \right)^{\delta} - \left( \frac{s^*(v_i, \Delta v_i)}{s_i} \right)^2 \right), \tag{26}$$

$$s^*(v_i, \Delta v_i) = s_0 + v_i T + \frac{v_i \Delta v_i}{2\sqrt{ab}}, \tag{27}$$

where $x_i$ and $v_i$ and are coordinate and velocity of $i$-th car, $a$ is a maximum acceleration, $\delta$ is acceleration exponent, $b$ is a vehicle deceleration, $s_0$ is a minimum gap between vehicles, $v_0$ is a desired speed, $T$ is a time gap between consecutive vehicles, $s_i$ is bumper-to-bumper distance between car $i$, and next riding car $i - 1$ expressed through the length $l_{i-1}$ of the car $i - 1$ as follows:

$$s_i = x_{i-1} - x_i - l_{i-1},$$

and $\Delta v_i$ is speed difference between speed of car $i$ and speed of car $i - 1$

$$\Delta v_i = v_i - v_{i-1}.$$

Typical values of the parameters $v_0$, $T$, $s_0$, $\delta$, $a$, $b$ are summarized in Table 1 taken from [19].

**Table 1.** Parameter values of intelligent driver model

| | |
|---|---|
| Desired speed $v_0$ | 54 km/h |
| Time gap $T$ | 1.0 s |
| Minimum gap $s_0$ | 2 m |
| Acceleration exponent $\delta$ | 4 |
| Acceleration $a$ | 1.0 m/s$^2$ |
| Comfortable deceleration $b$ | 1.5 m/s$^2$ |

In all algorithms based on deep Q-learning, we use the following neural network architecture. Since we want to speed up the learning process, it has only one hidden layer of size 64; the number of inputs equals 1, the number of outputs equals 3. We assume that the output size equals 3, since the network returns three values corresponding to the values of transmission at the next power level (which is one higher), the transmission at the current level, and transmission at the previous power level (which is one lower), respectively. In the case of dueling Q-learning, advantage and value layers are added to this architecture. We use a fairly low value of $\gamma = 0.7$, because the system is constantly changing and we want the network to concentrate more on current rewards than on the future rewards. Since the system is changing rapidly, we assume a low value for the *update_target_frequency* in dueling double Q-learning, so that the system can quickly adapt to new experiences. For the same reason, we assume a low value of *replay_mem_size* in all versions of deep Q-learning.

### 6.1. Single-Channel Game with Quadratic Power Function

In this section, we discuss simulations of a single-channel anti-jamming game. We assume that the vehicles are located on the same road, with the jammer chasing two communicating cars. Figures 3a and 4b show graphs of vehicle rewards and SINR in the case in which the distance between all consecutive network vehicles remains constant and equals 6.4 m. Figure 3a,b show graphs in the case when the initial vehicle coordinates equal 9.6 and 16 m, and the jammer initial coordinate is 0.8 m. In simulations corresponding to

Figure 4a,b, it is assumed that the vehicles are moving according to the intelligent driver model with the parameter values described in the previous section. Due to the fact that the car in front is moving according to the free road model (since there are no other vehicles in front of it), the distance between it and the transmitting vehicle increases over time, resulting in worse communication quality. This explains the decreasing of the graphs in Figure 4a,b.

The red color in all figures indicates the Nash equilibrium rewards and SINR calculated according to the formulas from the theorems proven in this article. As can be seen from Figures 3a and 4b, the graphs converge to theoretical predictions, confirming their correctness. We assume that the learning rate $\alpha$ is 0.05 in the case of PHC and 0.01 for the rest of the algorithms. We increase the learning rate, because otherwise the PHC would converge to the Nash equilibrium too slowly.

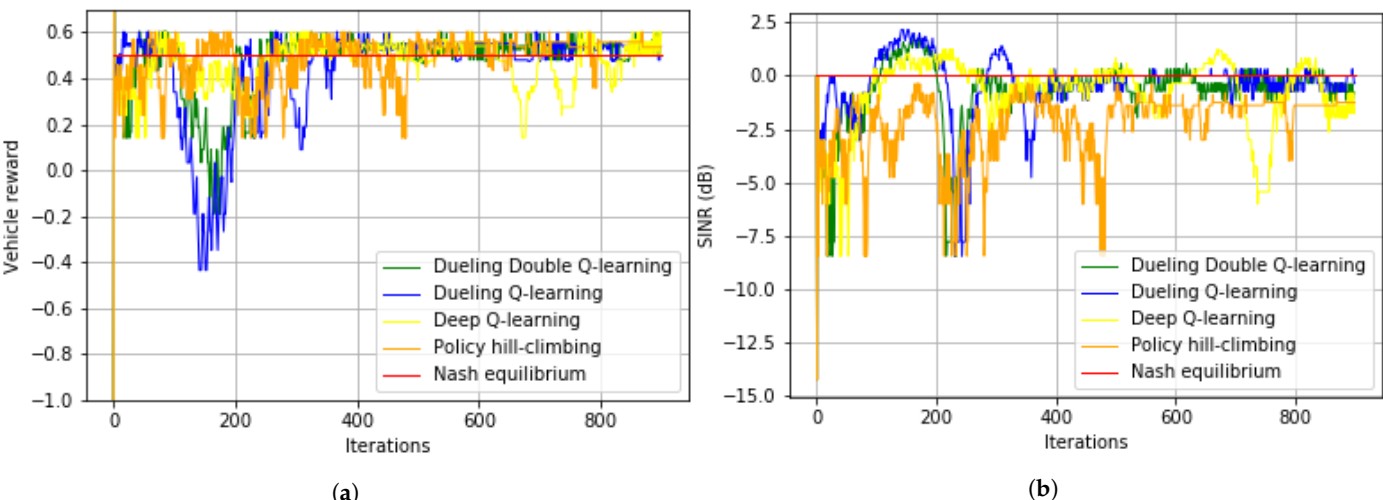

(**a**)                                    (**b**)

**Figure 3.** Single-channel game with quadratic power function and constant intervehicle distance: (**a**) vehicle rewards; (**b**) SINR.

Figure 3a,b show that dueling Q-learning and dueling double Q-learning are the most stable among all considered algorithms, while deep Q-learning and PHC have significant deviations from the Nash equilibrium. Figure 4a,b show that all the algorithms adapt quite well to changes in the network state; however, PHC is the closest to the Nash equilibrium throughout all iterations, while the rest of the algorithms deviates significantly from the optimal curve. This is due to the fact that Q-learning algorithms use experience replay, which allows for retraining using past experience. Even taking into account the fact that we chose the small buffer size *replay_mem_size* = 50 (usually such a buffer has a size of the order of 10,000), it can be seen from the simulations that such algorithms adapt to a change in environment with a noticeable delay; thus, if the state of the network changes rapidly, PHC is the best among considered algorithms.

It is worth noting that if we significantly increase the learning rate, this may result in the network being unable to converge to the optimal strategy due to the large amount of noise associated with the current state of the system. On the other hand, if the value of the learning rate is too low, then the learning will be too slow, which is especially bad in the case of a rapidly changing environment. Similar reasoning is valid for the size of the network. A significant increase in the size of the network will lead to a longer training process, which may be unacceptable in the case of online training. A smaller network may not be sophisticated enough to find the optimal strategy. The parameters that we use for modeling are obtained as a trade-off that is close to optimal.

### 6.2. Multi-Channel Game with Quadratic Power Function

Figures 5a and 6b show graphs in the case of a multi-channel game with $m = 3$ channels. We assume that in this case the vehicle changes the transmission channels

according to a predetermined pseudo-random sequence and each of the $m = 3$ channels in it is chosen with the same probability. Since the jammer must divide its power between channels in a multi-channel case, the rewards of the vehicle in this case are higher than in a single-channel case. To speed up the convergence rate, we increase the learning rate to 0.3 in the case of PHC and to 0.05 for other algorithms. Figure 5a,b show that all algorithms perform quite well in the case in which the distances between cars are constant, especially PHC and double dueling Q-learning; however, dueling Q-learning has a significant deviation from the Nash equilibrium at iterations 900–1000. In the dynamic case (intelligent driver model) presented in Figure 6a,b it can be seen that PHC adapts to a change in environment faster and deviates less from the instantaneous Nash equilibrium.

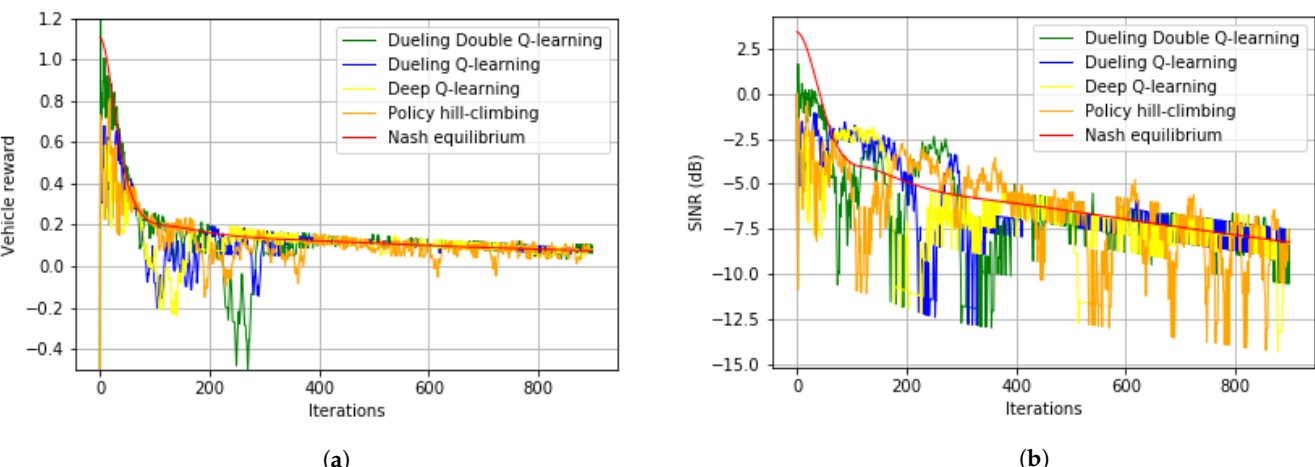

(**a**)　　　　　　　　　　　　　　　　(**b**)

**Figure 4.** Single-channel game with quadratic power function and variable intervehicle distance: (**a**) vehicle rewards; (**b**) SINR.

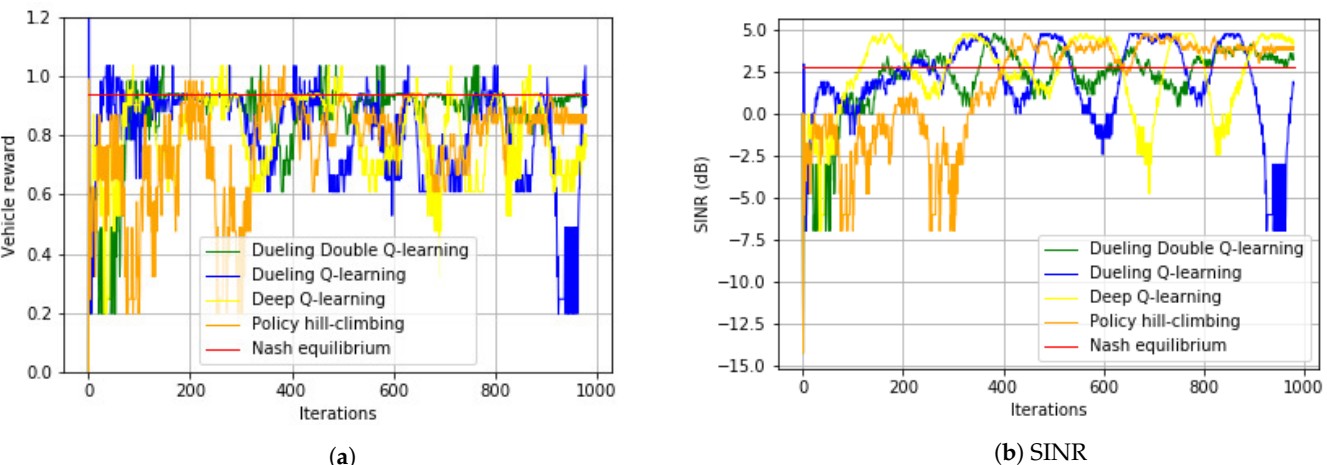

(**a**)　　　　　　　　　　　　　　　　(**b**) SINR

**Figure 5.** Multi-channel game with quadratic power function and constant intervehicle distance: (**a**) vehicle rewards; (**b**) SINR.

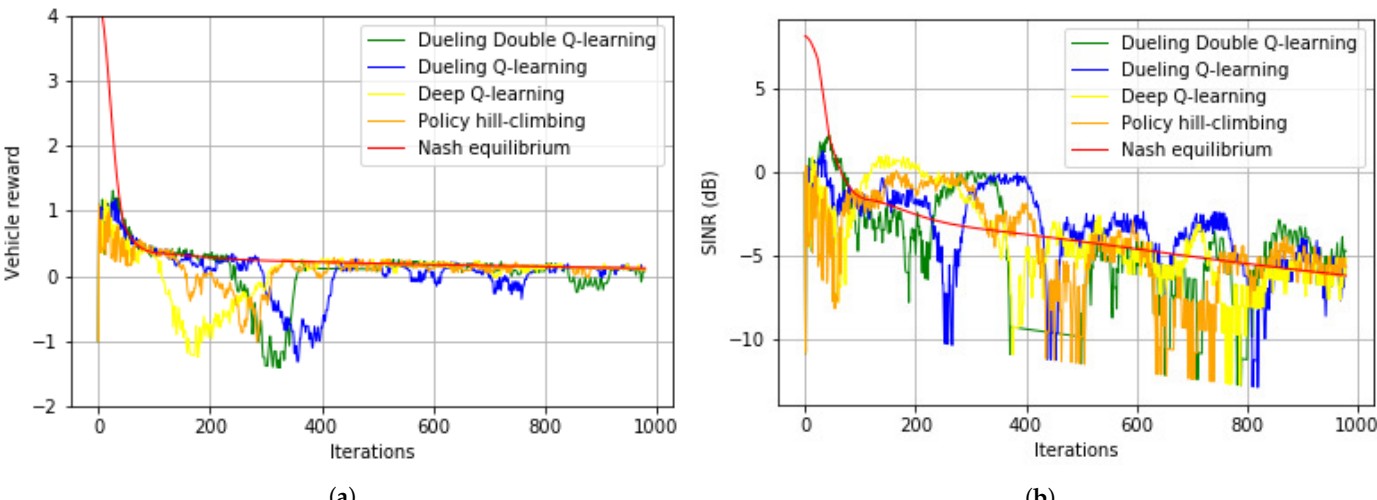

**Figure 6.** Multi-channel game with quadratic power function and variable intervehicle distance: (**a**) vehicle rewards; (**b**) SINR.

## 7. Summary

We considered a multi-channel VANET anti-jamming game with linear and quadratic power functions. This game is an antagonistic game between the jammer and a pair of communicating vehicles. At the beginning of the article, we assumed that the power term of the vehicle and the jammer utility functions is linear. In this case, we proved that the optimal strategy of the vehicle is signal transmission at the maximum power (see Theorem 1). We changed the utility functions by replacing the term linearly dependent on power with a quadratic one. We examined a single-channel and multi-channel game under this condition. In this case, the vehicle and the jammer strategies are not trivial and are better suited for practical implementation since the quadratic power term limits the growth of signal transmission power, and it remains at a more reasonable level. We expressed the Nash equilibrium of the system through communication parameters (Theorem 2). However, in practice, finding the Nash equilibrium can be problematic since communication parameters such as channel gains may not be known. Therefore, we considered modern machine learning algorithms such as deep Q-learning, dueling Q-learning, double dueling Q-learning, and policy hill climbing, and compared their performance. These algorithms, without having any information regarding the communication parameters, by trial and error, converge to the theoretically deduced Nash equilibrium; however, policy hill climbing shows better adaptability in the case of a rapidly changing system state.

The proposed model makes it possible to effectively combat jamming attacks under the condition that the vehicles are organized into a platoon. An interesting line of further research is the study of the case in which the signal transmission power term is a non-quadratic power function. It is also important to consider the model with unequal probabilities of channel jamming.

**Author Contributions:** Conceptualization, S.P. and G.D.; methodology, S.P. and G.D.; software, G.D.; validation, S.P. and G.D.; formal analysis, G.D.; investigation, G.D.; resources, G.D.; data curation, G.D.; writing—original draft preparation, S.P. and G.D.; writing—review and editing, S.P. and G.D.; visualization, G.D.; supervision, S.P.; project administration, S.P. and G.D.; funding acquisition, N/A. All authors have read and agreed to the published version of the manuscript.

**Funding:** This research received no external funding.

**Conflicts of Interest:** The authors declare no conflict of interest.

## Appendix A. Proof of Theorem 1

We first consider the case in which the system parameters satisfy the inequality

$$C_{car} \geq \frac{h_{car}^2}{\sigma^2}$$

In this case, the point $(0,0)$ is a Nash equilibrium, because

$$u_{car}(x,0) = x\left(\frac{h_{car}^2}{\sigma^2} - C_{car}\right) \leq 0 = u_{car}(0,0),$$

$$u_J(0,y) = -C_J y \leq 0 = u_{car}(0,0).$$

Thus, in this case, the optimal behavior of the vehicle and the jammer is to transmit zero power due to the high transmission cost $C_{car}$.

Let us consider the case

$$C_{car} < \frac{h_{car}^2}{\sigma^2}. \tag{A1}$$

The function $u_{car}$ is linear with respect to $x$, therefore, if we fix the value of $y$, it reaches a maximum at the ends of the segment $[0, P_C]$. Let us assume that the maximum is reached at the point $x^* = 0$. Since the function $u_J(x^*, y) = u_J(0, y) = -C_J y$ reaches its maximum at the point $y = 0$ we conclude that $y^* = 0$. However, it is impossible for the point $(0,0)$ to be a Nash equilibrium, since if $y^* = 0$ and inequality (A1) holds, then the function $u_{car}(x, 0)$ reaches its maximum at $x^* = P_C$, and not at $x^* = 0$. Thus, $x^* = P_C$.

Let us derive value of $y^*$. The derivative of utility function $u_{car}$ is given as follows:

$$\frac{d}{dy}u_J = \frac{d}{dy}\left(-\frac{h_{car}^2 x}{\sigma^2 + h_J^2 y} + C_{car}x - C_J y\right) = \frac{h_{car}^2 h_J^2 x}{(\sigma^2 + h_J^2 y)^2} - C_J. \tag{A2}$$

Solving the equation $\frac{d}{dy}u_J = 0$, we find its root $\hat{y}$

$$\hat{y} = \frac{-\sigma^2}{h_J^2} + \frac{h_{car}}{h_J}\sqrt{\frac{x}{C_J}}. \tag{A3}$$

Since $\frac{d^2}{dy^2}u_J < 0$, the function $u_J$ is convex downward and its maximum value can be reached at the point $\hat{y}$ or at the ends of the segment $0$ and $P_J$. Taking into account that the function $u_J$ increases with $y < \hat{y}$ and decreases with $y > \hat{y}$, we conclude that the maximum point $y^*$ can be calculated by Formula (5).

## Appendix B. Proof of Theorem 2

The proof consists of two parts. We first establish the equality $y_1^* = y_2^* = \ldots = y_m^*$. Denoting these variables by $y^*$, we reduce the problem to finding a Nash equilibrium in the two-dimensional case.

First, we obtain the derivatives $\frac{d}{dx}u_{car}$ and $\frac{d^2}{dx^2}u_{car}$

$$\frac{d}{dx}u_{car} = \sum_{k=1}^m \frac{d}{dx}\frac{1}{m}\left(\frac{h_{car}^2 x}{\sigma^2 + h_J^2 y_k} - C_{car}x^2\right) = \sum_{k=1}^m \frac{1}{m}\left(\frac{h_{car}^2}{\sigma^2 + h_J^2 y_k} - 2C_{car}x\right), \tag{A4}$$

$$\frac{d^2}{dx^2}u_{car} = -2C_{car} < 0. \tag{A5}$$

Let us consider the case $0 < x^* < P_C$ first. From (A5) we derive that the function $u_{car}(x, y_1^*, y_2^*, \ldots, y_m^*)$ is convex upward with respect to the variable $x$. Inequality (11) means that the point $x^*$ is a maximum of the function $u_{car}(x, y_1^*, y_2^*, \ldots, y_m^*)$. Therefore,

to fulfill inequality (11) it is necessary and sufficient that $\frac{d}{dx}u_{car}(x^*, y_1^*, y_2^*, \ldots, y_m^*) = 0$. Solving this equation, we find that

$$x^* = \frac{1}{2mC_{car}} \sum_{k=1}^{m} \frac{h_{car}^2}{\sigma^2 + h_J^2 y_k^*} \tag{A6}$$

Let us note that (12) is equivalent to the point $(y_1^*, y_2^*, \ldots, y_m^*)$ being the maximum of the function $u_J(x^*, y_1, y_2, \ldots, y_m)$. To find this maximum with restrictions (8), (9), and (A6) we consider the Lagrangian

$$L(x, y) = u_J - \lambda\left(\sum_k y_k - P_J\right) - \mu\left(\sum_{k=1}^{m} \frac{h_{car}^2}{\sigma^2 + h_J^2 y_k} - 2mC_{car}x\right),$$

where $\lambda \geq 0$. The derivative $\frac{d}{dy_l}L$ is calculated by the formula

$$\frac{d}{dy_l}L = \frac{d}{dy_l}u_J - \lambda + \mu\frac{h_{car}^2 h_J^2}{\left(\sigma^2 + h_J^2 y_l\right)^2} =$$

$$\sum_{k=1}^{m} \frac{1}{m}\frac{d}{dy_l}\left(-\frac{h_{car}^2 x}{\sigma^2 + h_J^2 y_k} + C_{car}x^2\right) - 2C_J y_l - \lambda + \mu\frac{h_{car}^2 h_J^2}{\left(\sigma^2 + h_J^2 y_l\right)^2} =$$

$$\frac{h_{car}^2 h_J^2 x}{m(\sigma^2 + h_J^2 y_l)^2} - 2C_J y_l - \lambda + \mu\frac{h_{car}^2 h_J^2}{\left(\sigma^2 + h_J^2 y_l\right)^2}. \tag{A7}$$

Solving the equation $\frac{d}{dy_l}L(x^*, y_1^*, y_2^*, \ldots, y_m^*) = 0$, we obtain

$$h_{car}^2 h_J^2 x^* + \mu m h_{car}^2 h_J^2 = m(2C_J y_l^* + \lambda)(\sigma^2 + h_J^2 y_l^*)^2. \tag{A8}$$

Let us note that Equation (A8) with respect to $y_l$ has no more than one root, since the left side is a constant, and the right side is a monotonically increasing function (because $\lambda \geq 0$). Let us denote this root by $y^*$. Since Equation (A8) is satisfied for every $l$, we conclude that

$$y_1^* = y_2^* = \ldots = y_m^* = y^*. \tag{A9}$$

The case $x^* = P_C$ can be analyzed in a similar way and also leads to Formula (A9). Thus, regardless of where $x^*$ is located on the interval $(0, P_C]$, equalities (A9) are satisfied.

Since (A9) holds, it is convenient for us to consider the functions $\tilde{u}_{car}(x, y)$ and $\tilde{u}_J(x, y)$ given as follows:

$$\tilde{u}_{car}(x, y) = u_{car}(x, y, y, \ldots, y) = \frac{h_{car}^2 x}{\sigma^2 + h_J^2 y} - C_{car}x^2, \tag{A10}$$

$$\tilde{u}_J(x, y) = u_J(x, y, y, \ldots, y) = -\frac{h_{car}^2 x}{\sigma^2 + h_J^2 y} + C_{car}x^2 - mC_J y^2. \tag{A11}$$

Let us establish that the function $u_J(x^*, y_1, y_2, \ldots, y_m)$ is a convex upwards function with regards to the variables $y_1, y_2, \ldots, y_m$. We rewrite the function $u_J(x^*, y_1, y_2, \ldots, y_m)$ in the following form:

$$u_J = \sum_{k=1}^{m} \frac{1}{m}\left(-\frac{h_{car}^2 x}{\sigma^2 + h_J^2 y_k} + C_{car}x^2 - mC_J y_k^2\right).$$

By proving that the second derivative is negative, it can be established that each term $-\frac{h_{car}^2 x}{\sigma^2 + h_J^2 y_k} + C_{car} x^2 - mC_J y_k^2$ is a convex upward function of the argument $y_k$. Therefore, each term is a convex upward function of the arguments $y_1, y_2, \ldots, y_m$. Thus, the function $u_J(x^*, y_1, y_2, \ldots, y_m)$ is a convex upward function of the arguments $y_1, y_2, \ldots, y_m$, since it equals to the sum of convex upward functions. Similarly, we can establish that the function $u_J(x, y_1^*, y_2^*, \ldots, y_m^*)$ is a convex upward function of the argument $x$.

From the convexity of the functions discussed in the previous paragraph we can conclude that the fulfillment of inequalities (11), (12) is equivalent to the fulfillment of the following inequalities:

$$\tilde{u}_{car}(x^*, y^*) \geq \tilde{u}_{car}(x, y^*), \tag{A12}$$

$$\tilde{u}_J(x^*, y^*) \geq \tilde{u}_J(x^*, y). \tag{A13}$$

Thus, we can reduce the multidimensional problem of finding a Nash equilibrium to the two-dimensional case. From (8) and (A9) we derive

$$0 < y^* \leq \frac{P_J}{m}.$$

Therefore, there are four different cases we need to consider: $(0 < x^* < P_C, 0 < y^* < \frac{P_J}{m})$, $(x^* = P_C, 0 < y^* < \frac{P_J}{m})$, $(0 < x^* < P_C, y^* = \frac{P_J}{m})$, and $(x^* = P_C, y^* = \frac{P_J}{m})$.

Case 1. $0 < x^* < P_C, 0 < y^* < \frac{P_J}{m}$. The derivatives $\frac{d}{dy}\tilde{u}_J(x, y)$ and $\frac{d^2}{dy^2}\tilde{u}_J(x, y)$ are given by the following formulas:

$$\frac{d}{dy}\tilde{u}_J(x, y) = \frac{h_{car}^2 h_J^2 x}{(\sigma^2 + h_J^2 y)^2} - 2mC_J y, \tag{A14}$$

$$\frac{d^2}{dy^2}\tilde{u}_J(x, y) = -\frac{2h_{car}^2 h_J^4 x}{(\sigma^2 + h_J^2 y)^3} - 2mC_J. \tag{A15}$$

From (A15) we conclude that

$$\frac{d^2}{dy^2}\tilde{u}_J(x, y) < 0. \tag{A16}$$

Therefore, the function $\tilde{u}_J$ of the argument $y$ is convex upward. Let us notice that (A13) is equivalent to the point $y^*$ being the maximum of the function $\tilde{u}_J(x^*, y)$. Therefore, the equality $\frac{d}{dy}\tilde{u}_J(x^*, y^*) = 0$ must be satisfied. By analogy, one can derive that

$$\frac{d^2}{dx^2}\tilde{u}_{car}(x, y) < 0 \tag{A17}$$

and conclude that the equality

$$\frac{d}{dx}\tilde{u}_J(x^*, y^*) = 0 \tag{A18}$$

must also hold.

From (A18) we obtain the following value of $x^*$:

$$x^* = \frac{h_{car}^2}{2C_{car}(\sigma^2 + h_J^2 y^*)}. \tag{A19}$$

Substituting (A19) into (A14) and equating $\frac{d}{dy}\tilde{u}_J(x^*, y^*)$ to zero we obtain

$$\frac{d}{dy}\tilde{u}_J(x^*, y^*) = \frac{h_{car}^4 h_J^2}{2C_{car}(\sigma^2 + h_J^2 y^*)^3} - 2mC_J y^* = 0,$$

$$y^*(\sigma^2 + h_J^2 y^*)^3 = \frac{h_{car}^4 h_J^2}{4m C_{car} C_J}.$$

Case 2. In the case $x^* = P_C$, $0 < y^* < \frac{P_J}{m}$ let us establish that the conditions $\frac{d}{dy}\tilde{u}_J(P_C, y^*) = 0$ and $\frac{d}{dx}\tilde{u}_{car}(P_C, y^*) \geq 0$ are necessary and sufficient for the point $(x^*, y^*)$ to be a Nash equilibrium.

We prove that fulfillment of the condition $\frac{d}{dx}\tilde{u}_{car}(P_C, y^*) \geq 0$ ensures that condition (A12) is satisfied. From (A17) we conclude that the derivative $\frac{d}{dx}\tilde{u}_{car}(x, y^*)$ decreases. If $\frac{d}{dx}\tilde{u}_{car}(x, y^*)$ is non-negative at the point $x = P_C$, then it is non-negative over the entire interval $0 < x \leq P_C$. Thus, in this interval, the function $\tilde{u}_{car}(x, y^*)$ is non-decreasing and condition (A12) is satisfied.

Let us establish that $\frac{d}{dy}\tilde{u}_J(P_C, y^*) = 0$ guarantees that (A13) holds. Since conditions $\frac{d^2}{dy^2}\tilde{u}_J(x, y) < 0$ and $\frac{d}{dy}\tilde{u}_J(P_C, y^*) = 0$ hold, we derive that $y^*$ is a maximum of the function $u_J(P_C, y)$. Therefore, (A13) is satisfied.

Carrying out the reverse reasoning, we can verify that the conditions

$$\frac{d}{dy}u_J(P_C, y^*) = 0 \tag{A20}$$

$$\frac{d}{dx}u_{car}(P_C, y^*) \geq 0 \tag{A21}$$

are also sufficient to satisfy (A12) and (A13).

From (A20) we obtain the following equality:

$$0 = \frac{d}{dy}\tilde{u}_J(P_C, y^*) = \frac{h_{car}^2 h_J^2 P_C}{(\sigma^2 + h_J^2 y^*)^2} - 2C_J m y^*,$$

which we can rewrite as

$$y^*(\sigma^2 + h_J^2 y^*)^2 = \frac{h_{car}^2 h_J^2 P_C}{2m C_J}. \tag{A22}$$

From (A21) we derive

$$\frac{d}{dx}\tilde{u}_{car}(P_C, y^*) = \frac{h_{car}^2}{\sigma^2 + h_J^2 y^*} - 2C_{car} P_C \geq 0.$$

Case 3. $0 < x^* < P_C$, $y^* = \frac{P_J}{m}$. By analogy with Case 2, we conclude that conditions $\frac{d}{dx}\tilde{u}_{car}(x^*, \frac{P_J}{m}) = 0$ and $\frac{d}{dy}\tilde{u}_J(x^*, \frac{P_J}{m}) \geq 0$ are necessary and sufficient for the point $(x^*, \frac{P_J}{m})$ to be a Nash equilibrium.

Therefore, we obtain the following equation:

$$0 = \frac{d}{dx}\tilde{u}_{car}\left(x^*, \frac{P_J}{m}\right) = \frac{h_{car}^2}{\sigma^2 + h_J^2 \frac{P_J}{m}} - 2C_{car} x^*.$$

From the last equation we conclude that

$$x^* = \frac{h_{car}^2}{2C_{car}(\sigma^2 + h_J^2 \frac{P_J}{m})}.$$

The inequality $\frac{d}{dy}\tilde{u}_J\left(x^*, \frac{P_J}{m}\right) \geq 0$ gives us

$$\frac{d}{dy}\tilde{u}_J\left(x^*, \frac{P_J}{m}\right) = \frac{h_{car}^2 h_J^2 x^*}{(\sigma^2 + h_J^2 \frac{P_J}{m})^2} - 2C_J P_J \geq 0.$$

Case 4. $x^* = P_C$, $y^* = \frac{P_J}{m}$. By analogy with Case 2, we conclude that the conditions $\frac{d}{dx}\tilde{u}_{car}\left(P_C, \frac{P_J}{m}\right) \geq 0$ and $\frac{d}{dy}\tilde{u}_J\left(P_C, \frac{P_J}{m}\right) \geq 0$ are necessary and sufficient for the point $\left(P_C, \frac{P_J}{m}\right)$ to be a Nash equilibrium. From the last two inequalities we deduce

$$\frac{d}{dx}\tilde{u}_{car}\left(P_C, \frac{P_J}{m}\right) = \frac{h_{car}^2}{\sigma^2 + h_J^2\frac{P_J}{m}} - 2C_{car}P_C \geq 0,$$

$$\frac{d}{dy}\tilde{u}_J\left(P_C, \frac{P_J}{m}\right) = \frac{h_{car}^2 h_J^2 P_C}{(\sigma^2 + h_J^2\frac{P_J}{m})^2} - 2C_J P_J \geq 0.$$

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
