# Peer review of "Jamming and Anti-Jamming Strategies of Mobile Vehicles"

_electronics, doi:10.3390/electronics10222772_

Round 1
Reviewer 1 Report
It would be beneficial to define "jam". To my understanding this refers only to the platoon case study, considering VANETs. Are there any other relative applications? All potential applications have to be referred, to strengthen the paper's potential. Also, which is the wireless protocol related to the paper's subject? As it is mentioned by the authors: "Anti-jamming games have become a popular research topic, however, there are not many publications devoted to such games in the case of Vehicular Ad-Hoc Networks (VANETs)." Why is that? Can the already established wireless protocols curate "jams" or not?
Clarification is needed for lines 17-18: "Within the framework of this theory, it is assumed that one or more cooperating vehicles maximize their utility functions, while each of the jammers maximizes its own." You need to provide an example of a utility function of each side. Which is the goal of each side?
Lines 250-252: "we prove that the optimal strategy of the vehicle is signal transmission at the maximum power". As the authors state, this is a trivial outcome. The actual contribution of the paper is summarized in lines 257-261, which is where the authors should focus and give more a detailed discussion.
The proposed changes should be pursued in order to evaluate the actual paper's potential.
References 5 and 6 are the same.
Author Response
Translator

Reviewer 2 Report
- English needs to be appropriate as a technical writing. For example, in line 6, it is written as “In addition, we compare the performance”….It could be either “we have compared…” or “we compared…”
- Abstract should be revised as lack of quantitative achievement is missing.
- Please add more recent relevant literatures in Introduction.
- The paper is well written, analysed and discussed. However, the novelty of this paper is not very clear. Would you please add the novelty of the work either in abstract or in conclusion?
- A block diagram or framework of complete work of this paper is required to add at the end of Chapter 1 so that readers can follow up the entire work has been done in this paper.
- Experimental work of anti-jamming game is missing. Please elaborate further on this.
- Performance/advantages comparison with existing related works (if available) would be good to add at the result section to validate the capability of the proposed method presented in the paper.
- Please add impact of this work in real life and future research direction.
Reviewer 3 Report
The abstract of this manuscript is well presented; however, it should be effectively reframed describing the background of intrusion detection initially, followed by the method adopted in the study and then the results and conclusion should be presented in quantitative form.
2. In section 1, it should be evident from the discussion that such a framework is actually needed? Part of the evidence could be resulting from past attempts or emphasizing the added value of such as reference framework. The rationale of the article should be clearly indicated in this section.
3. In section 2, explain what is the state-of-the-art and are there attempts towards developing such a framework that would better highlight the research challenge It could be the case that this is presented, but from reading the manuscript, especially the literature review, this is missing or unclear.
4. While, looking at the methods as presented, this is not seemingly the case and seems to come out as what the authors feel is the best direction and the underpinning or motivation from the state of the art is missing.
5. In section 6, can the author explain why specific parameters were taken. what happens if the parameter changes. Write and explain in a paragraph for support.
6. The result section is articulated effectively presenting the experimental results of the presented algorithm.
7. The conclusion section is well motivating but it is very lengthy. This section should clearly highlight the future research directions for the establishment of VANET.
Round 2
Reviewer 1 Report
The comments have been addressed. The current version of the paper is acceptable for publication.
Author Response
Thank you :)
Reviewer 2 Report
I was asking to add a block diagram of the technical work considered in this paper rather than putting the outline of the paper as a table format. Please revise this accordingly.
Author Response
Dear Reviewer,
Thank you very much for your valuable contribution. We have changed the diagram according to your feedback.
Sincerely yours,
Dr. Gleb Dubosarskii
Prof. Dr. Serguei Primak